# A dose-finding safety and feasibility study of oral activated charcoal and its effects on the gut microbiota in healthy volunteers not receiving antibiotics

**Armin Rashidi**[ID]<sup>1</sup>*, **Sathappan Karuppiah**[ID]<sup>2</sup>, **Maryam Ebadi**<sup>1</sup>, **Ryan Shanley**<sup>3</sup>, **Alexander Khoruts**<sup>4</sup>, **Daniel J. Weisdorf**<sup>1</sup>, **Christopher Staley**<sup>5</sup>

1 Division of Hematology, Oncology, and Transplantation, Department of Medicine, University of Minnesota, Minneapolis, MN, United States of America, 2 Department of Anesthesiology, University of Minnesota, Minneapolis, MN, United States of America, 3 Biostatistics Core, Masonic Cancer Center, University of Minnesota, Minneapolis, MN, United States of America, 4 Division of Gastroenterology, Hepatology, and Nutrition, Department of Medicine, University of Minnesota, Minneapolis, MN, United States of America, 5 Department of Surgery, University of Minnesota, Minneapolis, MN, United States of America

* arashidi@umn.edu

**Data Availability Statement:** All sequencing files are available from NCBI's Sequence Read Archive

## Abstract

Oral activated charcoal (OAC), a potent adsorbent with no systemic absorption, has been used for centuries to treat poisoning. Recent studies have suggested its potential efficacy in protecting the colonic microbiota against detrimental effects of antibiotics. In a dose-finding safety and feasibility clinical trial, 12 healthy volunteers not receiving antibiotics drank 4 different preparations made of 2 possible OAC doses (12 or 25 grams) mixed in 2 possible solutions (water or apple juice), 3 days a week for 2 weeks. Pre- and post-OAC stool samples underwent 16S rRNA gene sequencing and exact amplicon sequence variants were used to characterize the colonic microbiota. The preferred preparation was 12 grams of OAC in apple juice, with excellent safety and tolerability. OAC did not influence the gut microbiota in our healthy volunteers. These findings provide the critical preliminary data for future trials of OAC in patients receiving antibiotics.

## Introduction

Antibiotics represent a major cause of disruptions to the gut microbiota, leading to adverse consequences with global burden including *Clostridioides difficile* infection [1] and antibiotic resistance [2]. Some of the strategies that have been used to prevent antibiotic-related gut dysbiosis include judicious use of antibiotics, fecal microbiota transplantation, and selective luminal antibiotic degraders [3–5].

Activated charcoal is a potent adsorbent powder made from superheated, high-surface area, porous particles of organic material. The large surface area of activated charcoal is covered with a carbon-based network, allowing prompt adsorbance of chemicals. Because orally administered activated charcoal (OAC) is not systemically absorbed but can reduce the

database (https://www.ncbi.nlm.nih.gov/sra) with the accession number: SRP316695.

**Funding:** This work was supported by a University of Minnesota Medical School pilot award to A.R. and a National Institutes of Health's National Center for Advancing Translational Sciences (KL2TR002492). Conventional statistical analysis was performed with Biostatistics Shared Resource of the University of Minnesota Masonic Cancer Center, supported by NIH/NCI grant P30CA07759. The content is solely the responsibility of the authors and does not necessarily represent the official views of the National Institutes of Health's National Center for Advancing Translational Sciences. The funders had no role in study design, data collection and analysis, decision to publish, or preparation of the manuscript.

**Competing interests:** The authors have declared that no competing interests exist.

absorption of chemicals from the gastrointestinal lumen, it has been used for the treatment of poisoning for approximately two centuries [6]. As an over-the-counter product, OAC is also used for gas-related symptoms and a tooth whitener. Doses between 12 and 100 grams are frequently used in clinical practice. A slurry is produced by mixing the OAC power with water, sometimes flavored with juice to improve compliance. OAC-associated side effects are uncommon and are mostly limited to gastrointestinal symptoms such as fullness, nausea, and vomiting [6]. Two studies suggested that OAC may protect the gut microbiota by sequestering antibiotic residues in the lower gastrointestinal tract [7, 8].

The objectives of the present study were to determine (*i*) the preferred solution and a well-tolerated dose of OAC in healthy adults not receiving antibiotics and (*ii*) the effect of OAC on the gut microbiota in the absence of any medications. In the absence of prior data, we perceived these two pieces of knowledge to be the critical first steps towards planned clinical trials testing the efficacy of OAC in preventing antibiotic-related dysbiosis. In addition, although OAC is often used as an over-the-counter anti-gas treatment [9], its mechanism of action is unclear. We explored, via objective (*ii*), whether the reported anti-gas effect of OAC is mediated by specific microbiota changes.

## Materials and methods

### Clinical trial

We enrolled 12 healthy volunteers (8 men and 4 women) to a single-center, interventional, dose-finding protocol (registration number in ClinicalTrials.gov: NCT04204772; FDA IND: 143937; **Fig 1** and **S1 Appendix**). Study enrollment occurred in November 2020. Inclusion criteria were: age > 18 years, and no use of prescription medications within a month prior to consent. Exclusion criteria were: gastrointestinal symptoms within a month prior to consent, planned endoscopic procedure within a month after completing the study, known allergy to OAC, and sexually active women unwilling or unable to use non-oral forms of contraception. Eleven of these volunteers also consented to a local stool sample collection study (IRB approval number: STUDY00003519) where the donors could choose to provide one or more samples at arbitrary intervals. Both protocols were approved by the University of Minnesota IRB and ethics committee. All participants provided written informed consent.

OAC was purchased from Spectrum Chemical MFG Corp (New Brunswick, NJ; catalogue number CA131) and stored at room temperature. Microbial Limits Test by Pace Analytical Life Sciences, LLC (Oakdale, MN) determined that the total anaerobic microbial count and combined yeast and molds count were both <100 CFU/gram. Two doses (12 and 25 grams) and 2 solutions (water and commercially available apple juice) were used to make 4 possible preparations of charcoal. We designate these 4 preparations as W12 (12 grams of OAC in water), W25, J12 (12 grams of OAC in apple juice), and J25. The amount of OAC was predetermined, but participants could ask for more or less water or juice with each drink. Treatment assignments were predetermined using a balanced incomplete block design, where each participant received the same preparation in week 1 and switched to a different preparation in week 2. Participants presented to our facility in the morning of Monday, Tuesday, and Wednesday for 2 consecutive weeks, drank their OAC preparation, and were observed for 15 minutes after each drink. After each drink, the participants filled out a paper survey including a 5-point scale of their overall experience (scale: 1 = worst experience, 5 = best experience) since the previous OAC drink and a free-text field where the participants described their experience. Participants were contacted by phone on Thursday and Friday of weeks 1 and 2 and interim adverse events were recorded according to the Common Terminology Criteria for Adverse Events (CTCAE v.5.0). Participants received a $100 gift card as compensation on their last day of participation.

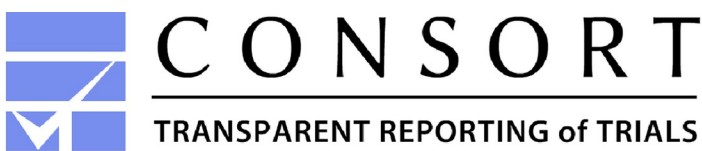

## CONSORT 2010 Flow Diagram

**Enrollment**

Assessed for eligibility (n=12)

Excluded (n=0)
- Not meeting inclusion criteria (n=0)
- Declined to participate (n=0)
- Other reasons (n=0)

Single arm (n=12)

**Allocation**

Allocated to intervention (n=12)
- Received allocated intervention (n=12)
    Week 1: W12, week 2: J25 (cohort 1; n=3)
    Week 1: W25, week 2: J12 (cohort 2; n=3)
    Week 1: J12, week 2: W25 (cohort 3; n=3)
    Week 1: J25, week 2: W12 (cohort 4; n=3)

**Follow-Up**

Lost to follow-up (n= 0)
Discontinued intervention (n=2)
    Throat discomfort (n=1, cohort 3)
    Dose measurement error (n=1; cohort 1)

**Analysis**

Analysed (n=12)
- Excluded from analysis (n= 1; cohort 1)
    1 subject (with dose measurement error)
    excluded only from analysis of preferred
    preparation

**Fig 1. CONSORT flow diagram.** Twelve healthy volunteers were enrolled in a single-arm interventional clinical trial. No subject was lost to follow up. One subject was taken off study after dose 3 because of dose measurement error; data from this subject were used for microbiome analysis, but not for determining the preferred charcoal preparation. Another subject discontinued participation after dose 5 because of throat discomfort. No subject was lost to follow up.

The participants who co-enrolled on both studies were asked to collect a stool sample on the weekend preceding dose 1 of OAC (pre-OAC sample) and another sample on the following weekend (prior to dose 4 of OAC; post-OAC sample). Stool samples were collected in 95% ethanol-filled sterile tubes and stored at -80˚C.

## Sequencing

DNA was extracted using the DNeasy PowerSoil DNA isolation kit (QIAGEN, Hilden, Germany). The V4 hypervariable region of the 16S rRNA gene was amplified on an Illumina MiSeq platform (2 x 300 paired-end mode) by the University of Minnesota Genomics Center [10]. Adaptor trimming was done in QIIME 2 using SHI7 [11] and the resulting demultiplexed fastq files were used as input to DADA2 [12]. Exact amplicon sequence variants (ASVs) were inferred from amplicon data using the *dada2* package v1.18.0 in R 3.4 (R Foundation for Statistical Computing, Vienna, Austria). For filtering, we used DADA2 default parameters (PHRED score threshold of 2, maximum number of expected errors of 2 for both forward and reverse reads) and truncation lengths of 240 and 160 for forward and reverse reads, respectively. De-replication, de-noising, merging, and chimera removal were done using DADA2 default parameters. Taxonomic assignment was done according to the naive Bayesian classifier method implemented within DADA2 and using the SILVA non-redundant v138.1 training set [13]. The ASV table was merged with clinical metadata into a phyloseq object for downstream analysis in R. Raw sequence reads were uploaded to the NCBI Sequence Read Archive and are accessible under BioProject ID SRP316695.

## Statistical analysis

We use notations pre-OAC and post-OAC to identify samples collected before dose 1 and at the end of week 1 (before dose 4), respectively. All analyses were performed in R using custom scripts and the following packages: *lme4*, *phyloseq*, *vegan*, and *aldex2*. A repeated measures analysis of variance (ANOVA) was done using OAC preparation as the fixed effect and subject number as a categorical random effect to determine whether ratings were different for different OAC preparations. Alpha diversity of the microbiota was estimated using the Shannon's $H$ index [14] and compared between pre- and post-OAC samples by a paired Wilcoxon signed-rank test. Beta diversity was estimated using the Aitchison distance and centered log-ratio (clr) abundances [15]. Ordination was visualized using principal component analysis of the distance matrix. An adonis test with 999 permutations was used to determine the partitioning of the distance matrix among sources of variation (subjects and pre- vs. post-OAC) [16]. Differential abundance analysis was performed on the genus-collapsed dataset using the aldex2 package in R, which used clr-transformed posterior distributions generated from 128 Dirichlet Monte-Carlo simulations [17]. Differentially abundant genera were identified by the aldex.ttest function in paired mode, with effect size determined by the aldex.effect function. Wilcoxon $p$ values from aldex.ttest corrected by the Benjamini-Hochberg method [18] ($q$ value threshold 0.10) and effect sizes from aldex.effect (threshold 1) were used to define differentially abundant genera in post- vs. pre-OAC samples.

## Results and discussion

The treatment was safe. One subject opted to come off study after dose 5 because of a transient uncomfortable feeling in her throat after each drink; she was not given the very last drink.

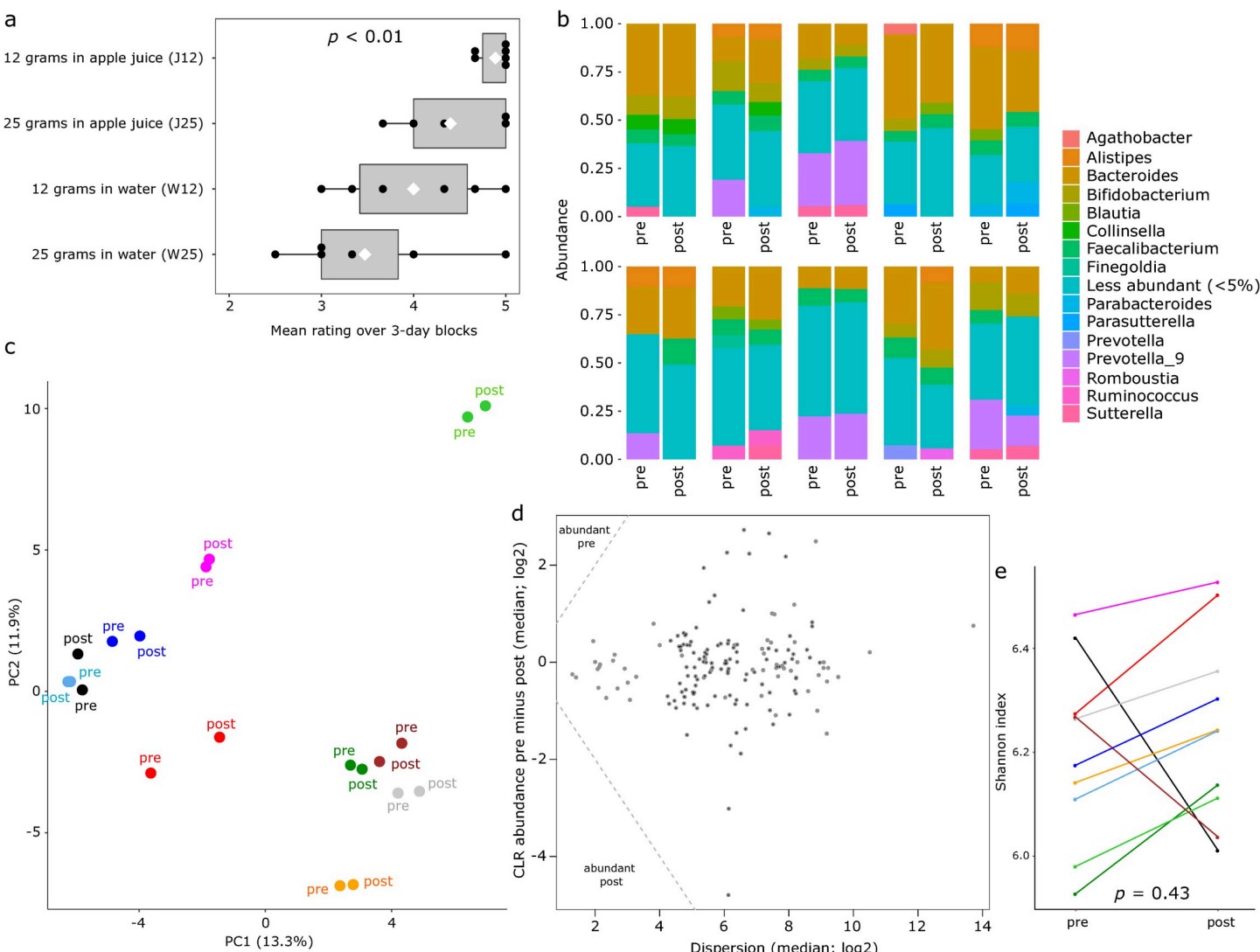

**Fig 2. Charcoal ratings and gut microbiota changes. (a)** Comparison between subject ratings of charcoal preparations. Each circle indicates the mean rating of a 3-day block by one subject. The white diamond indicates the mean of the 3-day block ratings for the corresponding charcoal preparation. The *p* value is from a repeated measures ANOVA, with a random subject factor. **(b)** Relative abundance of the most abundant genera in stool samples. Each bar represents one sample and a group of two bars represent pre- and post-charcoal samples from the same subject. Relative abundances are means across all samples and genera with an aggregate relative abundance of <5% are grouped together. **(c)** Principal components analysis using centered log-ratio abundances and Aitchison distance. Each circle represents a sample and circles in the same color represent pre- and post-charcoal samples from the same subject. PC1 and PC2 indicate the first and second principal components, with numbers in parentheses showing the fraction of microbiota variation explained by the corresponding axis. **(d)** *aldex2* output showing differentially abundant genera between pre- and post-charcoal samples. The triangles marked in left upper and left lower corners would contain differentially abundant genera. In this analysis, no genus was differentially abundant. An effect size threshold of 1 was used to define differential abundance. **(e)** Alpha diversity measured by Shannon index compared between pre- and post-charcoal samples in paired subject-specific mode. The *p* value is from a paired Wilcoxon signed-rank test.

Another subject (preparation W12) was taken off the study after dose 3 because of an OAC dose measurement error that did not allow ascertainment of the actual dose received. AEs were all grade 1 and included abdominal fullness/bloating in 4 (33%) subjects, abdominal pain in 4 (33%) subjects, nausea in 2 (17%) subjects, constipation in 2 (17%) subjects, and diarrhea in 1 (8%) subject. All subjects noticed transient black stools that resolved between weeks 1 and 2.

Subject rating of different OAC preparations is summarized in **Fig 2A**. Ratings were significantly different among the 4 groups (*p* < 0.01, repeated measures ANOVA). The OAC preparation with the highest rating was J12, with a mean (SD) of 4.9 (0.2) across 3-day blocks. This

was followed by J25 (mean rating 4.4, SD 0.6), W12 (mean rating 4.0, SD 0.8), and W25 (mean rating 3.5, SD 0.9). The 95% confidence interval for the mean difference relative to J12 was (-1.3, 0.2) for J25, (-1.6, -0.1) for W12, and (-2.0, -0.8) for W25. As these intervals are predominantly negative, with minimal to no overlap with zero, we concluded that J12 was the preferred preparation.

The pre-OAC sample from one subject had a disproportionately low depth (~3.5-fold lower than the next lowest depth sample) with poor coverage (~25%) and was deleted. Because our pre/post comparisons were paired, the post-OAC sample from the same subject was also deleted. This resulted in 20 samples with a median depth of 31,359 reads per sample (range: 21,561–85,586). 19,494 ASVs were assigned to 168 genera. The distribution of the most abundant genera among samples is shown in **Fig 2B**. Microbiota composition was highly subject-specific (explaining 76% of total microbiota variation; adonis $p < 0.001$), with little contribution by whether the sample was pre- vs. post-OAC (explaining only 2% of total microbiota variation, adonis $p = 0.33$)(**Fig 2C**). No genus was differentially abundant in pre- or post-OAC samples (**Fig 2D**). Alpha diversity did not change after OAC ($p = 0.43$; paired Wilcoxon signed-rank test; **Fig 2E**).

Together, these findings indicate no independent effect on the gut microbiome by two weeks (6 doses) of OAC. Therefore, gut microbiota changes after consuming OAC in patients receiving antibiotics in future trials would be due to the adsorbent effect of OAC on antibiotics rather than a direct effect on the microbiota. In addition, the often reported improvement in gas-related symptoms after using over-the-counter OAC by individuals not receiving antibiotics cannot be attributed to microbiota changes.

## Conclusions

In conclusion, we found 12 grams of OAC mixed with apple juice to be a suitable preparation of charcoal for future use in non-poisoning investigations. This preparation was safe and tolerable, and its administration was feasible. Apple juice mitigated the somewhat uncomfortable sensation that OAC created in the mouth and throat. This was likely the main reason for higher ratings obtained with apple juice than water. It is possible for other types of juice to have the same effect. We did not identify a direct effect by OAC on the gut microbiota in the absence of antibiotics.

In patients receiving intravenous antibiotics, OAC in the form of a solution as used in this study may be an easy, safe, and effective approach to protect the gut microbiome against the fraction of the antibiotics that reach the intestinal lumen (*e.g.*, via bile or direct transport). For patients receiving oral antibiotics, however, this strategy is not ideal because OAC can impair the absorption of antibiotics and reduce their desired systemic effect. Similarly, OAC would not be appropriate for patients on other essential oral medications that can be adsorbed by OAC. A potential patient who could benefit from OAC is someone requiring peri-procedural intravenous antibiotic prophylaxis for a non-gastrointestinal surgical procedure. *Clostridioides difficile* infection is a dysbiosis-related complication in these patients, and our findings here provide the critical preliminary data supporting a clinical trial of OAC in such patients [19, 20]. While we found a tolerable dose and palatable solution of OAC, therapeutic efficacy in patients may be higher at higher doses, and for patients receiving only one dose of OAC, palatability may not be as critical. The dose/solution identified in the present study should therefore be considered as a start point for future efficacy trials in patients.

## Supporting information

**S1 Checklist. TREND statement checklist.**
(PDF)

# PLOS ONE

Charcoal and the gut microbiota

**S1 Appendix. Clinical protocol.**
(DOCX)

# Acknowledgments

We thank Sharon Lopez for DNA extraction. Sequence data were processed and analyzed using the resources of the Minnesota Supercomputing Institute.

# Author Contributions

**Conceptualization:** Armin Rashidi, Alexander Khoruts.

**Data curation:** Armin Rashidi, Ryan Shanley.

**Formal analysis:** Armin Rashidi, Ryan Shanley, Christopher Staley.

**Funding acquisition:** Armin Rashidi.

**Investigation:** Armin Rashidi.

**Methodology:** Armin Rashidi, Ryan Shanley.

**Project administration:** Armin Rashidi, Sathappan Karuppiah, Maryam Ebadi.

**Resources:** Armin Rashidi.

**Supervision:** Armin Rashidi.

**Visualization:** Armin Rashidi, Ryan Shanley.

**Writing – original draft:** Armin Rashidi.

**Writing – review & editing:** Armin Rashidi, Sathappan Karuppiah, Maryam Ebadi, Ryan Shanley, Alexander Khoruts, Daniel J. Weisdorf, Christopher Staley.

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
