## [Decision Letter · Decision Letter 0]

25 Apr 2022

PONE-D-21-16930A Dose-Finding Safety and Feasibility Study of Oral Activated Charcoal and its Effects on the Gut Microbiota in Healthy Volunteers not Receiving AntibioticsPLOS ONE

Dear Dr. Rashidi,

Thank you for submitting your manuscript to PLOS ONE. After careful consideration, we feel that it has merit but does not fully meet PLOS ONE’s publication criteria as it currently stands. Therefore, we invite you to submit a revised version of the manuscript that addresses the points raised during the review process.

The manuscript has been evaluated by three reviewers, and their comments are available below.

The reviewers have raised a number of concerns that need attention. They request additional information on methodological aspects of the study to improve the quality of the reporting, as well as revisions to ensure the conclusions are presented appropriately.

Could you please revise the manuscript to carefully address the concerns raised?

We look forward to receiving your revised manuscript.

Kind regards,

Marianne Clemence

Staff Editor

PLOS ONE

Journal Requirements:

3. Thank you for stating the following financial disclosure: "This work was supported by a University of Minnesota Medical School pilot award to A.R. and a National Institutes of Health’s National Center for Advancing Translational Sciences (KL2TR002492). Conventional statistical analysis was performed with Biostatistics Shared Resource of the University of Minnesota Masonic Cancer Center, supported by NIH/NCI grant P30CA07759. The content is solely the responsibility of the authors and does not necessarily represent the official views of the National Institutes of Health’s National Center for Advancing Translational Sciences. The funders had no role in study design, data collection and analysis, decision to publish, or preparation of the manuscript."

We note that one or more of the authors is affiliated with the funding organization, indicating the funder may have had some role in the design, data collection, analysis or preparation of your manuscript for publication; in other words, the funder played an indirect role through the participation of the co-authors. If the funding organization did not play a role in the study design, data collection and analysis, decision to publish, or preparation of the manuscript and only provided financial support in the form of authors' salaries and/or research materials, please do the following:

a. Review your statements relating to the author contributions, and ensure you have specifically and accurately indicated the role(s) that these authors had in your study. These amendments should be made in the online form.

b. Confirm in your cover letter that you agree with the following statement, and we will change the online submission form on your behalf: 

“The funder provided support in the form of salaries for authors [insert relevant initials], but did not have any additional role in the study design, data collection and analysis, decision to publish, or preparation of the manuscript. The specific roles of these authors are articulated in the ‘author contributions’ section.

Reviewers' comments:

Reviewer's Responses to Questions

**Comments to the Author**

1. Is the manuscript technically sound, and do the data support the conclusions?

Reviewer #1: Yes

Reviewer #2: Yes

Reviewer #3: Yes

2. Has the statistical analysis been performed appropriately and rigorously? 

Reviewer #1: Yes

Reviewer #2: Yes

Reviewer #3: Yes

3. Have the authors made all data underlying the findings in their manuscript fully available?

Reviewer #1: Yes

Reviewer #2: Yes

Reviewer #3: Yes

4. Is the manuscript presented in an intelligible fashion and written in standard English?

Reviewer #1: Yes

Reviewer #2: Yes

Reviewer #3: Yes

5. Review Comments to the Author

Reviewer #1: A dose-finding safety and feasibility clinical trial was conducted in which 12 healthy volunteers drank 4 different preparations (2 OAC doses and 2 solutions). Pre- and post-OAC stool samples were gene sequenced. The preferred preparation was 12 gram of OAC in apple juice which demonstrated safety and tolerability.

Minor revisions:

1- Indicate the date range subjects were enrolled in the study.

2- Page 6: In addition to stating the frequencies of AEs, provide the corresponding percentages.

3- Indicate if AEs were collected according to standard methods, i.e. CTCAE.

Reviewer #2: This study nicely shows that OAC can be made palatable and does not appear to adversely affect the microbiome when dosed intermittently at low levels.

I agree that the low dose OAC with apple juice was liked by the most patients, but medicine doesn’t always have to be taste nice and an equally important question to be answered later is what is the correct dose for preventing antibiotic microbiome effects? It could be 12g, it could be 25g it could be more. What they actually showed is that lower dose OAC is more palatable and mixed with apple juice is more palatable. They This should be described in the results and acknowledged in the discussion. They also subsequently say the most likely scenario for this treatments use would be in surgical patients on IV antibiotics, where presumably the OAC therapy would only be needed 1-3 times at most. Maybe not so big a deal how nice it tastes.

The microbiome data is nice and very clearly shows that within an individual the OAC made no great difference. But this was in 12 people minus the dropout. Not really enough to say conclusively that OAC does not exert its anti-bloating effects by altering the microbiome.

Reviewer #3: PONE_21_16930 A Dose-Finding Safety and Feasibility Study of Oral Activated Charcoal and its Effects on the Gut Microbiota in Healthy Volunteers not Receiving Antibiotics

This is an interesting manuscript describing the dose and feasibility of OAC on the gut microbiota. It is not really an RCT given that only one arm is present and only before and after samples are used and compared within individuals. It is however, important information before the start of RCTs with antibiotics and OAC. I only have a few comments.

1. Consort diagram: would it be possible to add to the consort diagram the breakdown of people receiving the different amounts and the loss to follow-up for each of these?

2. It would be good to have a table with some of the characteristics for the participants e.g. gender, age, BMI. This could also be added as text.

3. The methods seem to suggest that the gut microbiota samples after were obtained prior to dose 4, the results suggest that the results were obtained after the two weeks (6 doses). Please check and make consistent.

4. Was the increased palatability of the OAC in apple juice due to masking of the taste? Would it therefore be possible to dissolve the OAC in other types of juices or other flavoured drinks as well?

5. How many doses do the authors think people should take when they receive IV prophylactic antibiotics in conjunction with procedures in order to protect the gut microbiota?

6. PLOS authors have the option to publish the peer review history of their article (what does this mean?). If published, this will include your full peer review and any attached files.

Reviewer #1: No

Reviewer #2: No

Reviewer #3: No

---

## [Author Response · Author response to Decision Letter 0]

27 Apr 2022

We thank the reviewers for their constructive feedback. Changes in the text are highlighted in yellow. 

Reviewer #1

Comment: Indicate the date range subjects were enrolled in the study.

Response: Enrollment occurred in Nov 2020. This was added to the revised manuscript.

Comment: Page 6: In addition to stating the frequencies of AEs, provide the corresponding percentages.

Response: This information was added to the revised manuscript.

Comment: Indicate if AEs were collected according to standard methods, i.e. CTCAE.

Response: AEs were collected according to CTCAE v.5.0. This is highlighted in the revised manuscript. 

Reviewer #2

Comment: This study nicely shows that OAC can be made palatable and does not appear to adversely affect the microbiome when dosed intermittently at low levels. I agree that the low dose OAC with apple juice was liked by the most patients, but medicine doesn’t always have to taste nice and an equally important question to be answered later is what is the correct dose for preventing antibiotic microbiome effects? It could be 12g, it could be 25g, it could be more. What they actually showed is that lower dose OAC is more palatable and mixed with apple juice is more palatable. This should be described in the results and acknowledged in the discussion. They also subsequently say the most likely scenario for this treatment's use would be in surgical patients on IV antibiotics, where presumably the OAC therapy would only be needed 1-3 times at most. Maybe not so big a deal how nice it tastes.

Response: We agree. To conduct the future trial in surgical patients, we needed a pilot study to get some idea about an appropriate dose/solution, understanding that it might not be the best choice. This information did not exist before this pilot study. Factors that could influence the optimal choice included palatability, potential side effects, feasibility, and compliance. However, as the reviewer correctly pointed out, a therapeutic effect may occur at various doses (not just the one we found as preferred among the tested options) and palatability of the solution may not be critical when consumed only once or a few times. We have added these to the revised manuscript (Introduction and Discussion) and changed “the preferred” to “a suitable” in the beginning of Discussion. 

Comment: The microbiome data is nice and very clearly shows that within an individual the OAC made no great difference. But this was in 12 people minus the dropout. Not really enough to say conclusively that OAC does not exert its anti-bloating effects by altering the microbiome.

Response: This is a valid point. All we can say is that this study did not identify an effect on the microbiome. We have adjusted the language in the revision. 

Reviewer #3

This is an interesting manuscript describing the dose and feasibility of OAC on the gut microbiota. It is not really an RCT given that only one arm is present and only before and after samples are used and compared within individuals. It is however, important information before the start of RCTs with antibiotics and OAC. I only have a few comments.

Comment: Consort diagram: would it be possible to add to the consort diagram the breakdown of people receiving the different amounts and the loss to follow-up for each of these?

Response: This information was added to the consent diagram. 

Comment: It would be good to have a table with some of the characteristics for the participants e.g. gender, age, BMI. This could also be added as text.

Response: Unfortunately, we did not collect age and BMI. Gender was added to the revised manuscript in the text. 

Comment: The methods seem to suggest that the gut microbiota samples after were obtained prior to dose 4, the results suggest that the results were obtained after the two weeks (6 doses). Please check and make consistent.

Response: Thank you for this clarifying question. Samples were indeed collected at baseline and after the first week (block 1), prior to dose 4. The study was designed like this because each subject went through 2 blocks, each with a different dose and solution. We would have not been able to attribute the information obtained from post-week 2 samples to week 1 vs. week 2 preparation, so we did not collect an end-of-study sample. The reason subjects went through block 2 was clinical (to assess AEs and tolerability). In the revised manuscript, we added a sentence in the beginning of Statistical Analysis which clarifies that our notation “post-OAC” refers to the pre-dose 4 sample. 

Comment: Was the increased palatability of the OAC in apple juice due to masking of the taste? Would it therefore be possible to dissolve the OAC in other types of juices or other flavoured drinks as well?

Response: OAC does not have any taste, but creates a somewhat uncomfortable sensation in the mouth/throat which was mitigated by dissolving it in the juice. We did not test other types of juice, but it is very possible that they would have the same effect as apple juice. We added this possibility in the revised manuscript (Conclusions, 1st paragraph). 

Comment: How many doses do the authors think people should take when they receive IV prophylactic antibiotics in conjunction with procedures in order to protect the gut microbiota?

Response: This is a very good question that only a trial in patients would be able to answer with certainty. We speculate that 1 dose would be enough but this definitely needs to be tested in a trial. To avoid conveying a premature message to the readers, we did not make a speculation in the manuscript. In the last paragraph of the revised manuscript, we briefly discuss this.

---

## [Decision Letter · Decision Letter 1]

2 Jun 2022

A Dose-Finding Safety and Feasibility Study of Oral Activated Charcoal and its Effects on the Gut Microbiota in Healthy Volunteers not Receiving Antibiotics

PONE-D-21-16930R1

Dear Dr. Rashidi,

We’re pleased to inform you that your manuscript has been judged scientifically suitable for publication and will be formally accepted for publication once it meets all outstanding technical requirements.

Kind regards,

James Mockridge

Staff Editor

PLOS ONE

Reviewers' comments:

Reviewer's Responses to Questions

**Comments to the Author**

1. If the authors have adequately addressed your comments raised in a previous round of review and you feel that this manuscript is now acceptable for publication, you may indicate that here to bypass the “Comments to the Author” section, enter your conflict of interest statement in the “Confidential to Editor” section, and submit your "Accept" recommendation.

Reviewer #1: All comments have been addressed

Reviewer #2: All comments have been addressed

Reviewer #3: All comments have been addressed

2. Is the manuscript technically sound, and do the data support the conclusions?

Reviewer #1: (No Response)

Reviewer #2: Yes

Reviewer #3: (No Response)

3. Has the statistical analysis been performed appropriately and rigorously? 

Reviewer #1: (No Response)

Reviewer #2: Yes

Reviewer #3: (No Response)

4. Have the authors made all data underlying the findings in their manuscript fully available?

Reviewer #1: (No Response)

Reviewer #2: Yes

Reviewer #3: (No Response)

5. Is the manuscript presented in an intelligible fashion and written in standard English?

Reviewer #1: (No Response)

Reviewer #2: Yes

Reviewer #3: (No Response)

6. Review Comments to the Author

Reviewer #1: (No Response)

Reviewer #2: The authors have listened to the reviewer comments and have answered them all to my satisfaction noting the limitations of the study.

Reviewer #3: (No Response)

7. PLOS authors have the option to publish the peer review history of their article (what does this mean?). If published, this will include your full peer review and any attached files.

Reviewer #1: No

Reviewer #2: No

Reviewer #3: No

---

## [Editor Report · Acceptance letter]

6 Jun 2022

PONE-D-21-16930R1 

A Dose-Finding Safety and Feasibility Study of Oral Activated Charcoal and its Effects on the Gut Microbiota in Healthy Volunteers not Receiving Antibiotics 

Dear Dr. Rashidi:

I'm pleased to inform you that your manuscript has been deemed suitable for publication in PLOS ONE. Congratulations! Your manuscript is now with our production department. 

Kind regards, 

on behalf of

Dr James Mockridge 

Staff Editor

PLOS ONE